# Aerodynamic Effects of Knitted Wire Meshes—CFD Simulations of the Flow Field and Influence on the Flow Separation of a Backward-Facing Ramp

**Jan Hauke Harmening [1],\*, Harish Devananthan [1], Franz-Josef Peitzmann [1] and Bettar Ould el Moctar [2]**

[1] Department of Mechanical Engineering, Mechatronics Institute Bocholt, Westphalian University, Münsterstraße 265, 46397 Bocholt, Germany

[2] Department of Mechanical Engineering, Institute for Ship Technology, Ocean Engineering and Transport Systems, University Duisburg-Essen, Bismarckstraße 69, 47057 Duisburg, Germany

\* Correspondence: jan.harmening@w-hs.de

**Abstract:** Passive flow control techniques are needed to reduce flow separation and enhance aerodynamic performance. In this work, the effect of a knitted wire mesh on the flow separation of a backward-facing ramp was numerically investigated for a Reynolds number of 3000. A grid independence study and a RANS turbulence model sensitivity analysis were conducted. The CFD simulations exhibited counter-rotating streamwise vortices emerging from the knitted wire mesh, and the reattachment length was significantly reduced. A variation of the knitted wire rows revealed a maximum reduction of the reattachment length of 25.7% for the case of four rows. A comparison with a different knitted wire mesh geometry yielded a decreased reattachment length reduction.

**Keywords:** knitted wire mesh; backward-facing ramp; verification and validation; reattachment length; streamwise vortices

## 1. Introduction

Reduction of flow separation is crucial to decrease drag. The abundant research on this topic can be divided into active and passive flow control. Active control mechanisms require an external energy supply and comprise fluidic, plasma, and moving surface actuators [1,2]. Passive flow control mechanisms affect the flow field via geometric adjustments, including tubercles, vortex generators, dimples, and tripping wires [3–8].

Several investigations have shown positive effects of a straight tripping wire on flow separation. Son et al. investigated a tripping wire on the surface of a sphere for different Reynolds numbers and noted a reduction in drag of more than 60% [7]. Yadegari and Khoshnevis conducted simulations and measurements of a tripping wire positioned on the surface of an elliptic cylinder and reported drag reductions of up to 75% [9]. Choudhry et al. experimentally investigated effects of an elevated wire positioned close to the surface of the NACA 0012 airfoil. They reported counter-rotating spanwise-oriented vortices and a delay of the separation [10]. These drag reduction capabilities of straight wires were confirmed by other investigations [11,12] and can be attributed to a tripping of the laminar boundary layer into a turbulent boundary layer [7]. This enhances the mixing of the fluid, transferring higher momentum fluid close to the wall and thus increasing resistance against an adverse pressure gradient that causes flow separation.

Multiple wires in a lattice structure can have a beneficial effect on the flow separation as well. In a recent study, Pelacci et al. demonstrated the drag reduction of a cylinder using a woven lattice structure [13]. The authors described significant drag reductions of up to 45%. Similar effects are known for permeable surfaces that reduce friction due to the permeability of the coating material [14,15].

Aside from woven meshes, wires can also be knitted. A knitted wire mesh consists of several lines of curved wires that form intersecting stitches. This is a crucial difference

from woven wire meshes, where the wires are aligned in a grid of nearly straight lines [13]. The work presented above does not cover single curved wires nor wire stitches that are aggregated in a knitted structure. Hence, the effect of knitted wire mesh structures on flow separation is an open question.

The research presented above indicates that single knitted wires introduce vorticity into the flow, which enhances the mixing and therefore reduces flow separation. Unlike straight tripping wires, the curved knitted structure introduces three-dimensional vorticity into the flow. Additionally, a knitted wire mesh consisting of multiple wires may reduce flow separation. However, the existence of a beneficial effect of knitted wire meshes has not been investigated yet.

The shape of the stitches of a knitted wire mesh resembles the edges of a dimple. Dimples are circular indentations of the surface that are known to reduce the flow separation and drag of spheres and airfoils [6,16]. The drag reduction capabilities can be attributed to streamwise vorticity stemming from the flow over their curved edges [17,18]. Hence, the shape of the stitches might also introduce streamwise vortices that have a beneficial impact on flow separation.

Some research has already been done concerning knitted structures in flows. Several publications have investigated the pressure drop or permeability of knitted structures [19–21]. The pressure drop is an important question in filter technology, where the streamlines of the flow are oriented normal to the filter plane. Typically, unstructured or structured computational grids are used for close-up studies of unit-cells of the knitted structure [19–21]. Other research has covered the question of a mathematical description of the knitted wire course [20,22]. Further work has been done on the geometric simplification of knitted meshes [23].

The literature review presented above shows that small scale investigations have been conducted for fluids flowing through unit-cells of the knitted structure. To the best knowledge of the authors' knowledge, neither detailed simulations nor measurements of knitted wire structures positioned on a flat plate have yet been published.

In our work, steady state CFD simulations of the flow over knitted wire meshes placed on a flat plate upstream of a backward-facing ramp are presented. The scope of our study consists of comparing the effect on the flow field and its separation of a knitted wire mesh, arranged in a manner consisting of between one and five rows, as well as for a second kind of knitted wire mesh. Furthermore, we analyze the numerical accuracy of the results as well as the sensitivity of different turbulence models.

## 2. Setup

For the numerical investigations, two different kinds of knitted wire meshes were used. In this work, we investigated the 0.28 mm diameter (type A) and the 0.7 mm diameter (type B) variants of Eloona GmbH (Pleinfeld, Germany). Both knitted structures differed in the shape of the stitches, with type A being flatter than type B (see Figure 1).

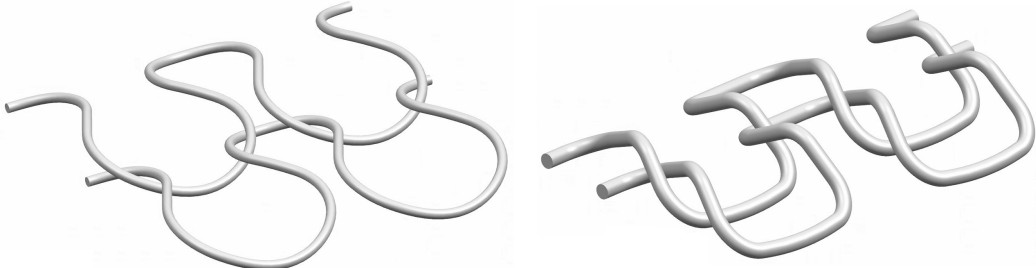

**Figure 1.** Wire meshes under investigation. **Left**: type A, **right**: type B. A total of four stitches distributed in two rows are shown for each type.

Idealized and parameterized CAD models of the knitted wire meshes were used for the analysis. Table 1 lists the geometric parameters of the knitted wire meshes. Both types

of knitted wires were scaled to a diameter of 1 mm to focus the investigations on the effect of the wire geometry.

**Table 1.** Geometric parameters of the knitted wire meshes.

| Parameter | Type A | Type B |
|---|---|---|
| Wire diameter | 0.28 mm | 0.7 mm |
| Stitch width | 5.35 mm | 10 mm |
| Stitch length | 4.7 mm | 4 mm |
| Knitted wire mesh thickness h | 0.95 mm | 3.62 mm |

For the numerical investigations, the wire mesh was modeled on the bottom of a channel upstream of a backward-facing ramp (see Figure 2). Table 2 lists the geometric parameters of the channel. The span S of the channel comprises two stitches. This gives a ratio $S/H_2$ of 3.8 for the type A and 2.8 for type B wire mesh. The edges of the ramp are blended with a radius of $r/H_2 = 0.5$.

The Reynolds number Re based on the step height was 3000. This Re was selected since it was shown to be well within the range where the reattachment length is minimal and nearly independent of Re [24,25].

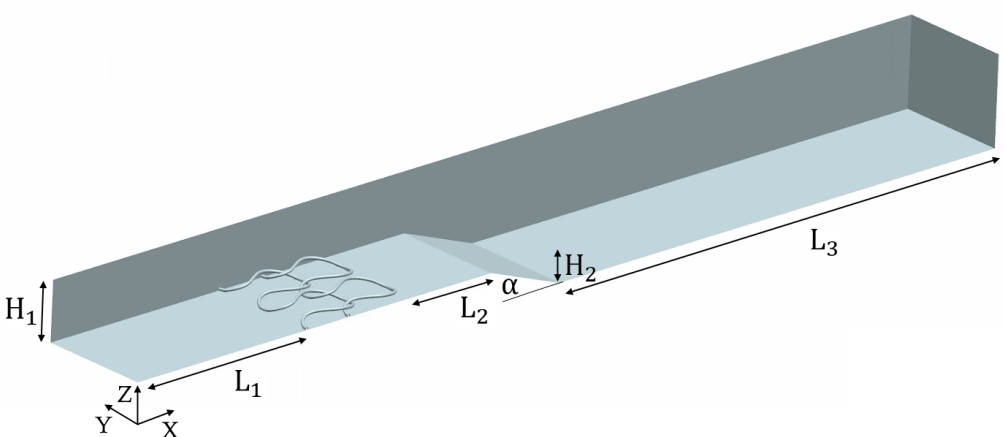

**Figure 2.** Geometry of the backward-facing ramp including the knitted wire mesh.

**Table 2.** Geometrical parameters of the backward-facing ramp channel.

| Parameter | Measure |
|---|---|
| $H_1$ | $2\,H_2$ |
| $H_2$ | 10 mm |
| $L_1$ | $6.66\,H_2$ |
| $L_2$ | $3.2\,H_2$ |
| $L_3$ | $20\,H_2$ |
| $\alpha$ | $20°$ |

For the analysis, two planes were used to extract and visualize the results, R and S, where R represents the symmetry plane between two stitches and S represents the symmetry plane of a single stitch. The results were extracted along several vertical lines, R1 to R5 and S1 to S5. Figure 3 displays both planes oriented normal to the channel bottom, as well as the positions of the lines. Table 3 lists the positions of the line probes shown in Figure 3. The lines R1 to R5 are positioned on the symmetry plane R between two stitches, and the lines S1 to S5 are positioned on the symmetry plane S of one stitch. Lines R3 and S3 are positioned at the center of the backward-facing ramp.

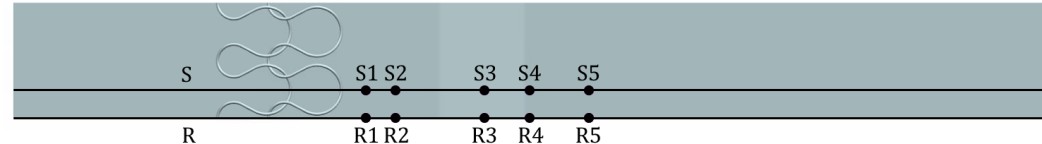

**Figure 3.** Line probes S1 to S5 and R1 to R5 on the planes S and R used to evaluate the results. The lines indicate the planes that are oriented normal to the channel bottom.

**Table 3.** Coordinates of the line probes.

|        | R1    | R2    | R3    | R4   | R5   | S1    | S2    | S3    | S4   | S5   |
|--------|-------|-------|-------|------|------|-------|-------|-------|------|------|
| $X/H_2$ | 11.5  | 12.5  | 15.49 | 17.0 | 19.0 | 11.5  | 12.5  | 15.49 | 17.0 | 19.0 |
| $Y/H_2$ | 0     | 0     | 0     | 0    | 0    | 2.86  | 2.86  | 2.86  | 2.86 | 2.86 |

To analyze discretization and modeling errors, a grid sensitivity and turbulence modeling error analysis was conducted using the type A knitted wire mesh with two rows. Furthermore, a validation was performed with direct numerical simulation (DNS) data for a separated flow around a backward-facing step. To investigate the effect of the number of rows, type A was used with a row number ranging from one to five. To test the effect of a different knitted wire mesh geometry, type B with two rows was analyzed.

## 3. Numerical Method

All simulations were carried out with the commercial finite volume CFD software STAR-CCM+. For all models, the three-dimensional incompressible Reynolds averaged Navier–Stokes (RANS) equations were solved. The stationary RANS equations including the Boussinesq hypothesis:

$$\frac{\partial \overline{u_i}}{\partial x_i} = 0 \tag{1}$$

$$\overline{u_j}\frac{\partial \overline{u_i}}{\partial x_j} + \frac{\partial}{\partial x_j}\left(\nu_t\left(\frac{\partial \overline{u_i}}{\partial x_j} + \frac{\partial \overline{u_j}}{\partial x_i}\right) - \frac{2}{3}k\delta_{ij}\right) = -\frac{1}{\rho}\frac{\partial \overline{p}}{\partial x_i} + \nu\frac{\partial^2 \overline{u_i}}{\partial x_j \partial x_i} \tag{2}$$

with mean velocity $\overline{u}$, mean pressure $\overline{p}$, density $\rho$, kinematic viscosity $\nu$ and turbulent kinetic energy $k$. The fluid was considered as air with a density $\rho$ of 1.18 kg/m³ and a viscosity $\mu$ of $18.55 \times 10^{-6}$. Equations (1) and (2) are the continuity equation and the momentum equation, respectively. To calculate the turbulent viscosity $\nu_t$, the $k$-$\omega$ SST model of Menter [26], the $k$-$\omega$ model of Wilcox [27], the $\gamma$-$Re_\theta$-$k$-$\omega$ SST model of Menter et al. [28], the two-layer realizable $k$-$\varepsilon$ model of Rodi [29], the standard low Re $k$-$\varepsilon$ turbulence model of Lien et al. [30] and the one-equation SA model by Spalart and Allmaras [31] were used. All turbulence models were low Reynolds models. For further information on the turbulence models, the reader is referred to the STAR-CCM+ user manual.

## 4. Numerical Setup

The whole domain was meshed using the trimmed cell-meshing algorithm, resulting in a computational grid consisting of mostly hexahedral cells. On the bottom of the channel as well as on the wire surface, a prism layer mesh with 13 prism layers was deployed. The thickness of the first layer was set to 0.02 mm. Where the wires touched the channel ground, the edges were blended with a radius of $r/H_2 = 0.012$ to facilitate the prism layer meshing of the knitted wires. Surrounding the knitted wire mesh and downstream of the backward-facing ramp, the grid was refined by a factor of two. The $y^+$ values were below unity at all locations for all simulations. All applied grids satisfied the grid quality acceptance limits for volume change, skewness angle and face validity of 0.01, 85° and 1.0, respectively. Figures 4 and 5 show the computational grid deployed. An isotropic mesh sizing was defined for the grid. Hence, the spanwise grid was of the same structure as shown in Figures 4 and 5.

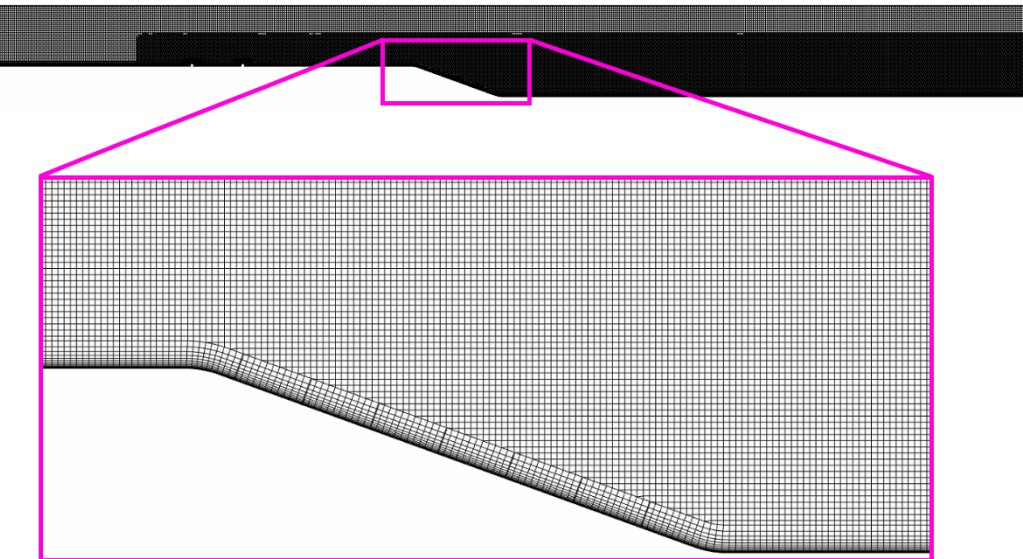

**Figure 4.** Computational grid surrounding the ramp on plane R.

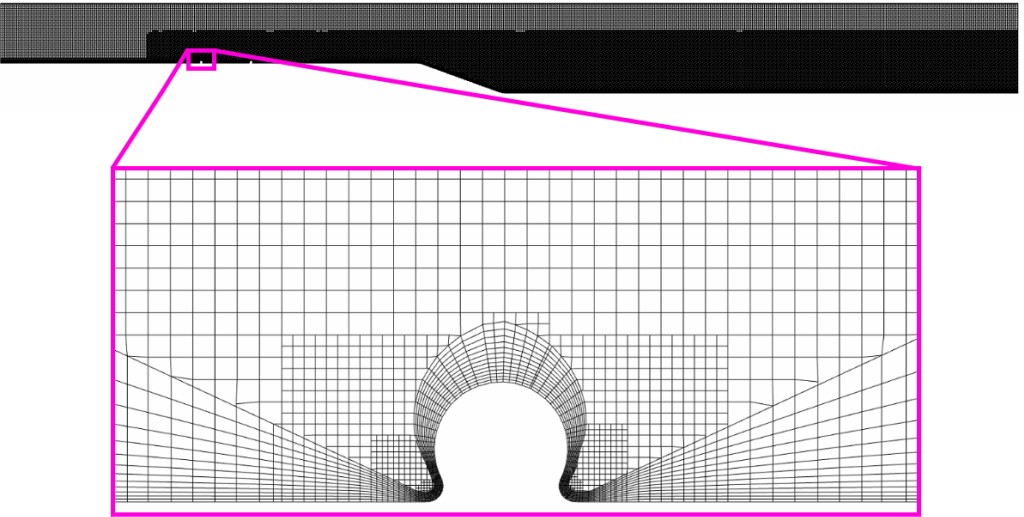

**Figure 5.** Computational grid surrounding the knitted wire mesh on plane R.

For all turbulence models, a hybrid wall function was selected that blended the logarithmic boundary layer profile of the outer layer and the linear profile in the viscous sublayer. The blending was performed according to Reichhardt's law [32]. All calculations were run using the second-order upwind discretization scheme. Pressure–velocity coupling was achieved with the SIMPLE algorithm. All simulations were iterated until the residuals converged to the minimal achievable iteration error.

For the channel bottom and the wire surface, a no-slip wall was specified. For all other surfaces, a symmetry plane was defined. A pressure outlet was deployed at the outlet. At the inlet of the channel, a constant velocity of 4.64 m/s with a turbulence intensity of 0.01 and a turbulent viscosity ratio of 10 was applied. The turbulence intensity I was defined as $I = \sqrt{2k/3}/\overline{u}$ and the turbulent viscosity ratio as $\mu_t/\mu$.

From the ERCOFTAC database [33], Yang and Voke provided records for a large eddy simulation (LES) of a developing boundary layer on a flat plate with an upstream turbulence intensity of 5% [34]. Figure 6 displays these comparative LES results together with a boundary layer that was obtained using the numerical method described above. The profiles were determined at X = 51.7 mm, i.e., upstream of the knitted wire mesh at a Reynolds number of 15,300. Yang and Voke also supplied profiles of the associated

root-mean-square velocity fluctuations. Figure 6 displays these profiles, together with the isotropic fluctuations based on the RANS simulations extracted from the turbulent kinetic energy. The RANS simulation yielded a slightly thicker boundary layer than the LES. The fluctuations extracted from the RANS calculation differed from the LES results due to their isotropic nature. Furthermore, a lower level of free-stream turbulence was predicted by the RANS simulations. The boundary layer profiles were found to be in acceptable agreement. The ratio of the knitted wire mesh height h to the boundary layer thickness $\delta$, obtained at $L_1$, gave $h/\delta = 2$ for type A and $h/\delta = 3$ for type B. This is in agreement with the ratio $h/\delta = 2.5$ used in an experimental analysis by Baldacchino et al. [35].

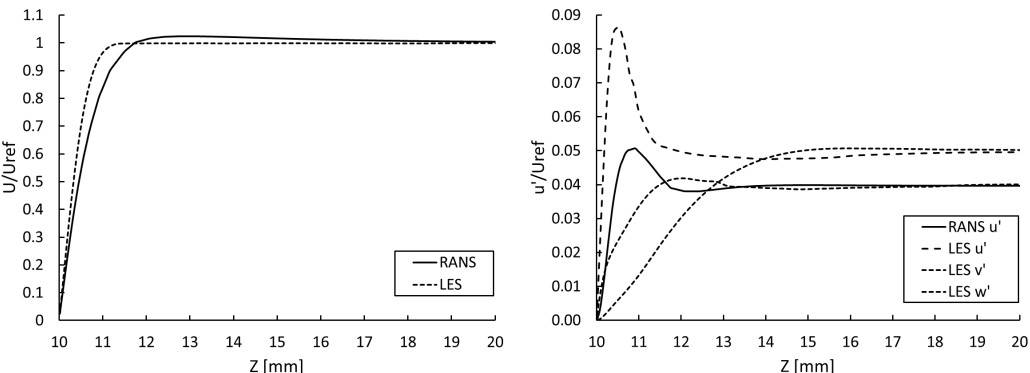

**Figure 6.** Boundary layer profile in comparison with LES data [34] for an upstream turbulence intensity of 5%. **Left**: normalized axial velocity, **right**: normalized root-mean-square velocity fluctuations.

## 5. Verification and Validation

To assess numerical and modeling errors, a grid sensitivity analysis as well as a turbulence model sensitivity analysis were conducted. Additionally, the accuracy of the numerical method was validated with data from a direct numerical simulation (DNS).

### 5.1. Grid Sensitivity Analysis

The grid sensitivity analysis was conducted for the type A knitted wire mesh with two rows of wires. The discretization error and the associated uncertainty was assessed on several vertical lines and estimated using the least squares method of Eça and Hoekstra [36]. The cases of convergence and divergence were distinguished using the convergence ratio

$$R = \frac{\phi_3 - \phi_2}{\phi_2 - \phi_1} \tag{3}$$

with the solutions $\phi_3$, $\phi_2$ and $\phi_1$ on the constant refined computational grids using the SST turbulence model. The convergence and divergence criteria were distinguished as follows:

- Monotonic convergence: $0 < R < 1$;
- Oscillatory convergence: $-1 < R < 0$;
- Monotonic divergence: $R > 1$;
- Oscillatory divergence: $R < -1$.

In some areas, divergence was detected using the convergence ratio criterion, although the solution already had converged. This was the case for the free stream above the separated flow, where almost constant velocity values were calculated for all grids. In this case, no discretization error could be estimated formally. To facilitate statements about the discretization error, the least squares method was applied in case the deviations between the results on the different grids were below 0.5%. All other divergent nodes were excluded from the analysis.

Three grids were created to estimate the discretization error. The refinement factor was 1.5. Figures 4 and 5 display the medium grid consisting of more than ten million cells.

The results on the three grids were evaluated along the lines S1 to S5 and R1 to R5 located in the wake of the knitted wire mesh and the separated flow over the ramp.

Calculations with the $k$-$\omega$ SST model converged to residuals below $2 \times 10^{-6}$ for all equations except for the turbulent kinetic energy $k$, which was minimized to a residual on the order of $10^{-5}$ for the medium grid.

As an example, Figures 7 and 8 display the velocity results obtained on the three computational grids for the line probes R4 and S4, which were positioned at the half span of the ramp.

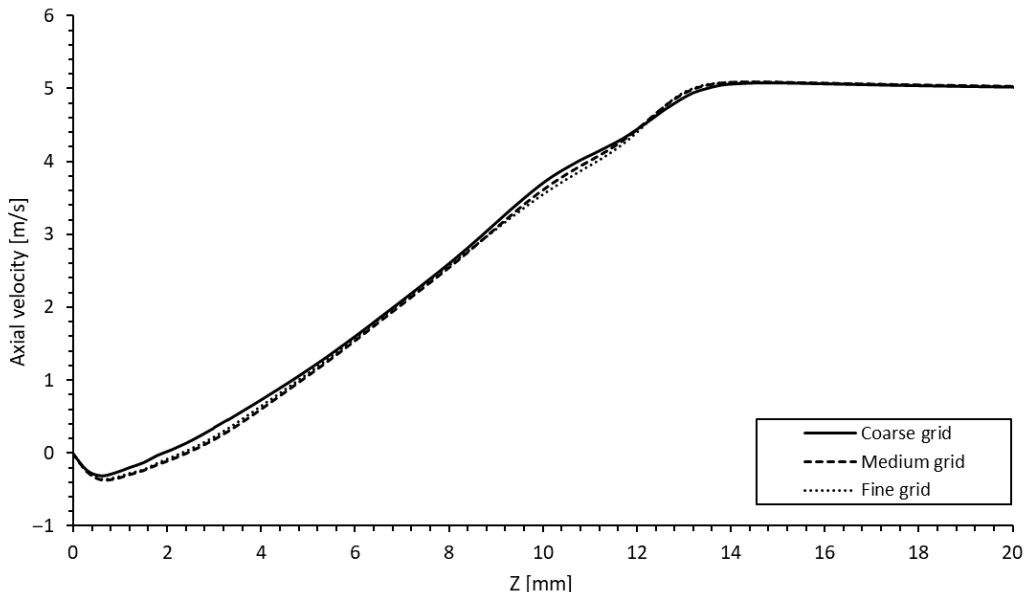

**Figure 7.** Simulated axial velocity for the three grids at R4. The ordinate is clipped at Z = 20.

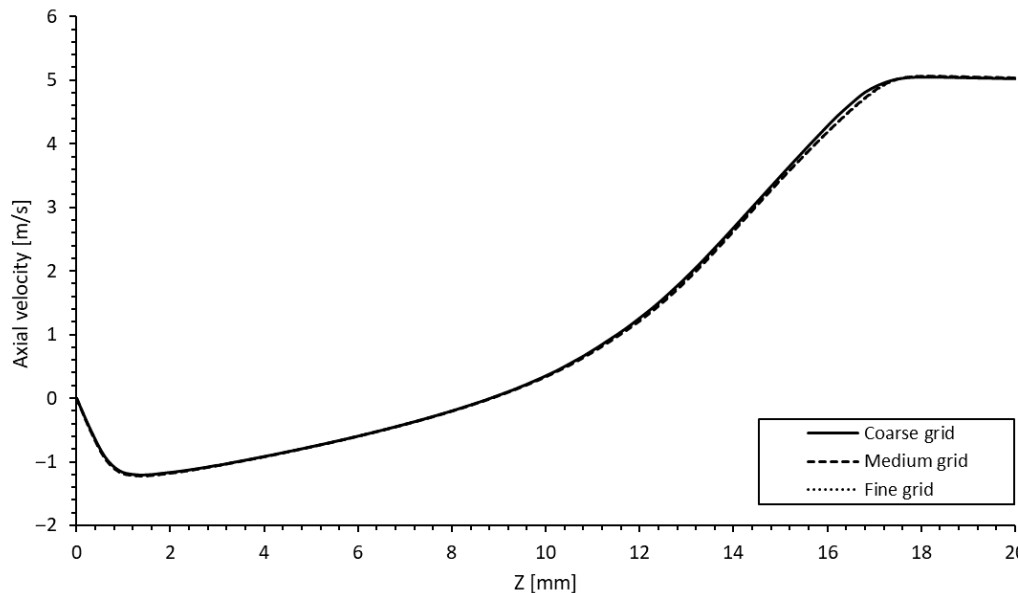

**Figure 8.** Simulated axial velocity for the three grids at S4. The ordinate is clipped at Z = 20.

As seen, the difference between the solutions decreased with grid refinement. While the differences between the solutions for the coarse and the medium grid are distinct, the solutions for the medium and the fine grid are almost indistinguishable.

Table 4 lists the mean relative absolute error (MRAE), the normalized mean squared error (NMSE) and the validation metric V of Oberkampf and Trucano [37] between the

grids for all vertical lines investigated. While the global MRAE between the coarse and the medium grid was 4%, the global MRAE for the medium and fine grid decreased to 1.6%.

**Table 4.** Verification metrics for results on the applied computational grids.

| Line Probe | Metrics: Coarse vs. Medium | | | Metrics: Medium vs. Fine | | |
|---|---|---|---|---|---|---|
| | MRAE | NMSE | V | MRAE | NMSE | V |
| R1 | 0.0065 | $1.4 \times 10^{-4}$ | 0.99 | 0.0032 | $2.3 \times 10^{-5}$ | 1.00 |
| R2 | 0.0052 | $7.7 \times 10^{-5}$ | 0.99 | 0.0019 | $7.2 \times 10^{-6}$ | 1.00 |
| R3 | 0.0104 | $4.6 \times 10^{-5}$ | 0.99 | 0.0107 | $5.8 \times 10^{-5}$ | 0.99 |
| R4 | 0.1022 | $2.6 \times 10^{-4}$ | 0.93 | 0.0476 | $4.4 \times 10^{-5}$ | 0.97 |
| R5 | 0.0109 | $2.2 \times 10^{-5}$ | 0.99 | 0.0139 | $5.8 \times 10^{-5}$ | 0.99 |
| S1 | 0.0405 | $1.8 \times 10^{-3}$ | 0.96 | 0.0286 | $1.3 \times 10^{-4}$ | 0.97 |
| S2 | 0.1702 | $5.3 \times 10^{-3}$ | 0.91 | 0.0260 | $7.4 \times 10^{-5}$ | 0.97 |
| S3 | 0.0268 | $2.0 \times 10^{-4}$ | 0.97 | 0.0057 | $5.4 \times 10^{-6}$ | 0.99 |
| S4 | 0.0091 | $9.2 \times 10^{-5}$ | 0.99 | 0.0059 | $1.3 \times 10^{-5}$ | 0.99 |
| S5 | 0.0243 | $3.7 \times 10^{-5}$ | 0.98 | 0.0112 | $1.3 \times 10^{-5}$ | 0.99 |
| Mean | 0.0406 | $8.0 \times 10^{-4}$ | 0.97 | 0.0155 | $4.2 \times 10^{-5}$ | 0.99 |

Table 5 presents the error metrics between the medium grid solution and the estimated grid-independent solution. The estimated mean relative absolute discretization error for the medium grid was 2.0% with a mean uncertainty U of 0.016 m/s. Consequently, the turbulence model sensitivity analysis was expected to be independent of a significant discretization error. This was emphasized by the results for the NMSE and V that indicated excellent global agreement between the estimated grid-independent solution and the medium grid solution.

**Table 5.** Verification metrics for results on the medium grid and the estimated exact solution.

| Line Probe | Metrics: Medium vs. Grid-Independent | | | |
|---|---|---|---|---|
| | MRAE | NMSE | V | U |
| R1 | 0.0082 | $1.7 \times 10^{-4}$ | 0.99 | −0.0847 |
| R2 | 0.0066 | $6.1 \times 10^{-5}$ | 0.99 | −0.0552 |
| R3 | 0.0173 | $1.0 \times 10^{-4}$ | 0.98 | −0.0136 |
| R4 | 0.0607 | $2.7 \times 10^{-4}$ | 0.97 | 0.0346 |
| R5 | 0.0064 | $6.9 \times 10^{-5}$ | 0.99 | −0.0141 |
| S1 | 0.0273 | $4.9 \times 10^{-4}$ | 0.97 | 0.0462 |
| S2 | 0.0448 | $1.3 \times 10^{-3}$ | 0.96 | 0.2211 |
| S3 | 0.0075 | $3.8 \times 10^{-5}$ | 0.99 | 0.0210 |
| S4 | 0.0050 | $1.6 \times 10^{-5}$ | 1.00 | 0.0111 |
| S5 | 0.0225 | $4.1 \times 10^{-5}$ | 0.98 | −0.0094 |
| Mean | 0.0206 | $2.5 \times 10^{-4}$ | 0.98 | 0.0157 |

*5.2. Turbulence Model Sensitivity Analysis*

The turbulence model sensitivity analysis was conducted using the *k*-*ω* SST, the *k*-*ω*, the *γ*-*Re*$_θ$-*k*-*ω* SST, the two-layer realizable *k*-*ε*, the standard low Re *k*-*ε* and the SA models. as described above. The models were assessed on the vertical lines R1 to R5 and S1 to S5. As examples, Figures 9 and 10 display the results of the five models on the medium grid at R4 and S4. The realizable *k*-*ε* model yielded different solutions than the other models. All other models varied partially, but yielded qualitatively similar results.

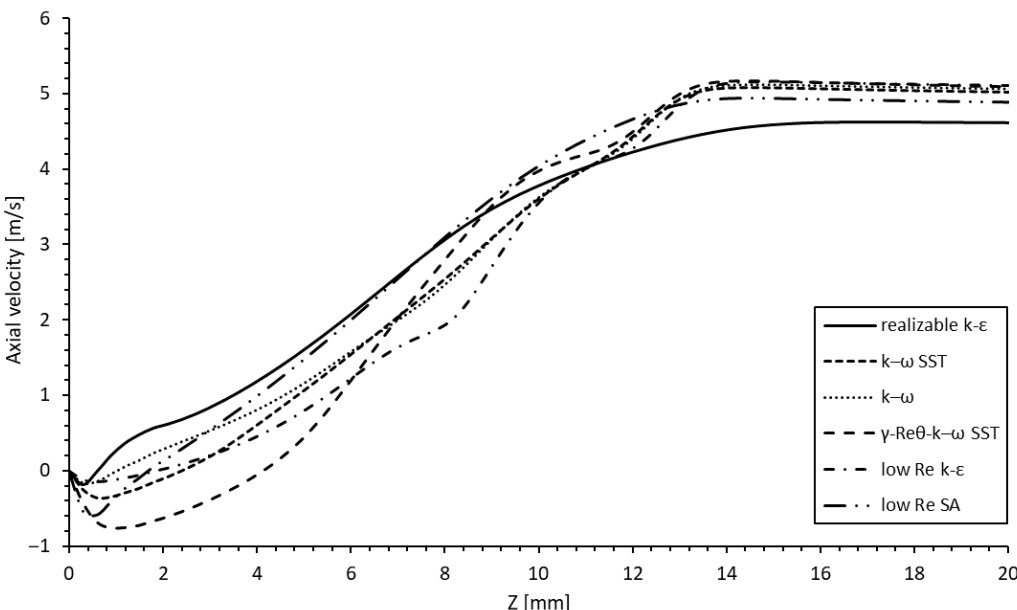

**Figure 9.** Simulated axial velocity magnitude at R4 for different turbulence models. The ordinate is clipped at Z = 20.

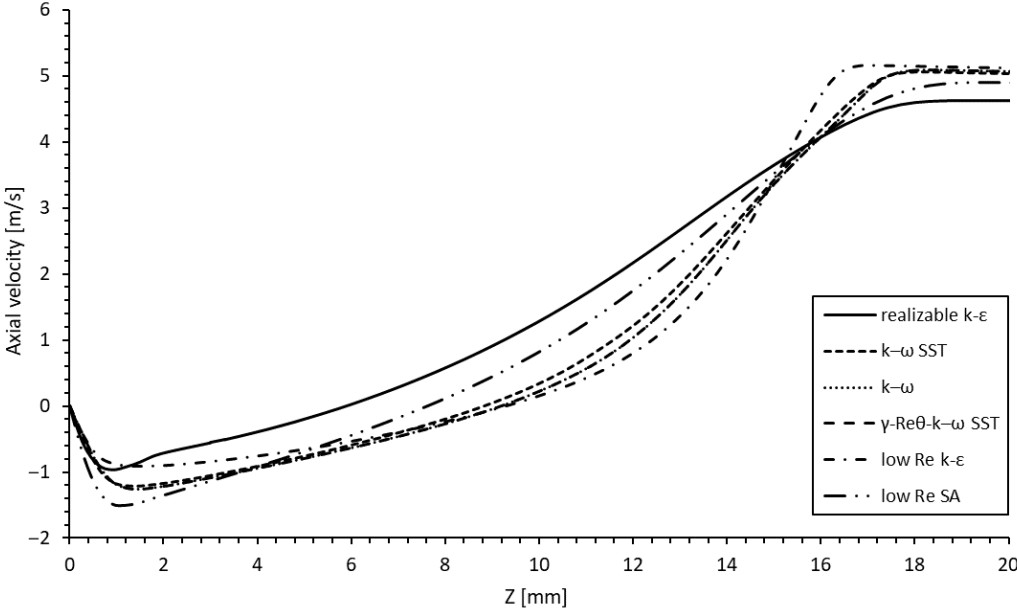

**Figure 10.** Simulated axial velocity magnitude at S4 for different turbulence models. The ordinate is clipped at Z = 20.

This is also visible in Figure 11, which exhibits the contour plot of the simulated velocity magnitudes around the backward-facing step for the low Reynolds *k-ε*, the *k-ω* SST and the *γ-Re*$_θ$-*k-ω* SST model at the symmetry plane R in between the stitches. The models predicted a small recirculation area at the upper edge of the ramp and a separated area at the downstream end of the ramp. While the models varied in the shape of the separated flow areas, they predicted qualitatively similar flow fields.

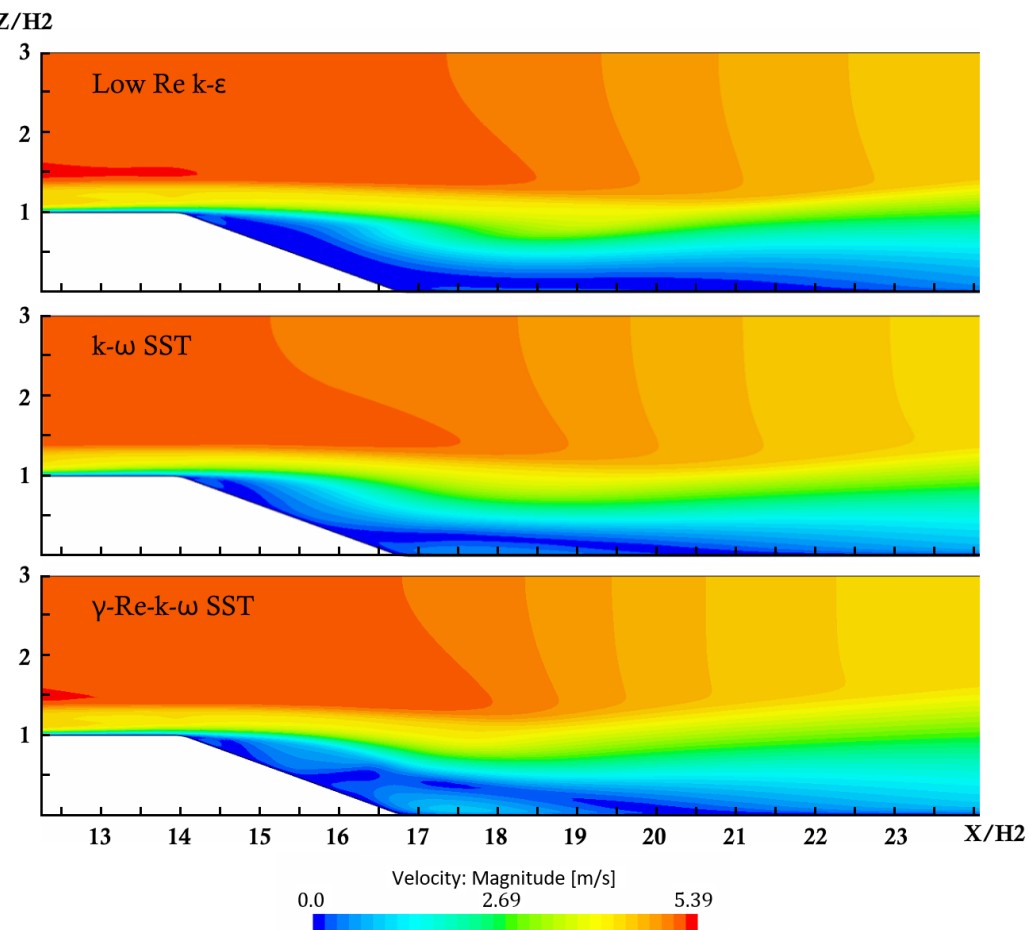

**Figure 11.** Simulated velocity magnitude profiles around the backward-facing ramp for the type A knitted wire mesh with two rows using different turbulence models.

Table 6 lists the calculated reattachment length for the backward-facing ramp with and without the knitted wire mesh for the turbulence models. The reattachment length was defined as the distance from the downstream end of the ramp to the reattachment. The simulated reductions varied between 9.2% for the realizable *k-ε* model and 21.2% for the *γ-Re*$_θ$*-k-ω* SST model. The mean reattachment reduction of all models, excluding the realizable *k-ε* model, was 17.4%. The *k-ω* SST model predicted a reduction of 19.0% and, hence, it is suitably representative.

**Table 6.** Calculated reattachment lengths for different turbulence models.

| Turbulence Model | Reattachment Length Clean in mm | Mean Reattachment Length Knitted Wire Mesh in mm | Mean Reduction |
|---|---|---|---|
| *k-ω* SST | 72.3 | 58.5 | 19.0% |
| Standard *k-ω* | 77.5 | 67.9 | 12.3% |
| *γ-Re*$_θ$*-k-ω* SST | 62.0 | 48.9 | 21.2% |
| Low Re *k-ε* | 74.1 | 59.0 | 20.4% |
| Realizable *k-ε* | 34.5 | 31.3 | 9.2% |
| Low Re SA | 62.4 | 53.6 | 14.2% |

The turbulence model sensitivity analysis showed that the results of the *k-ω* SST model agreed well with several other RANS turbulence models. All models predicted a significant reduction of the reattachment length. All models except the realizable *k-ε* model predicted a recirculation area on the upstream edge of the ramp and a separation bubble on the

downstream end of the ramp on the symmetry plane R in between the stitches. Hence, the *k-ω* SST model was expected to simulate the qualitative flow field correctly and, thus, it was used for all further investigations.

### 5.3. Validation with DNS Data

To analyze the capability of the *k-ω* SST model for capturing flow separation on the computational grid, we conducted a validation study using the DNS data of Le and Moin [38] for the flow around a backward-facing step. The DNS data were obtained for the flow at a Reynolds number of 5100, which is close to the Reynolds number of 3000 of our flow. For validation, the backward-facing step geometry was modeled with the same method described in the Numerical Setup. The benchmark DNS data were taken from the ERCOFTAC data base [33].

Figure 12 displays the vertical distribution of axial velocity along several lines downstream of the step. The *k-ω* SST model predicted an elongated separation bubble as well as a smaller and steeper shear layer downstream of the reattachment. Overall, the RANS results agreed favorably with the DNS data. This finding is in accordance with several numerical investigations of a backward-facing step that reported favorable agreements between the model and experiments [39–41].

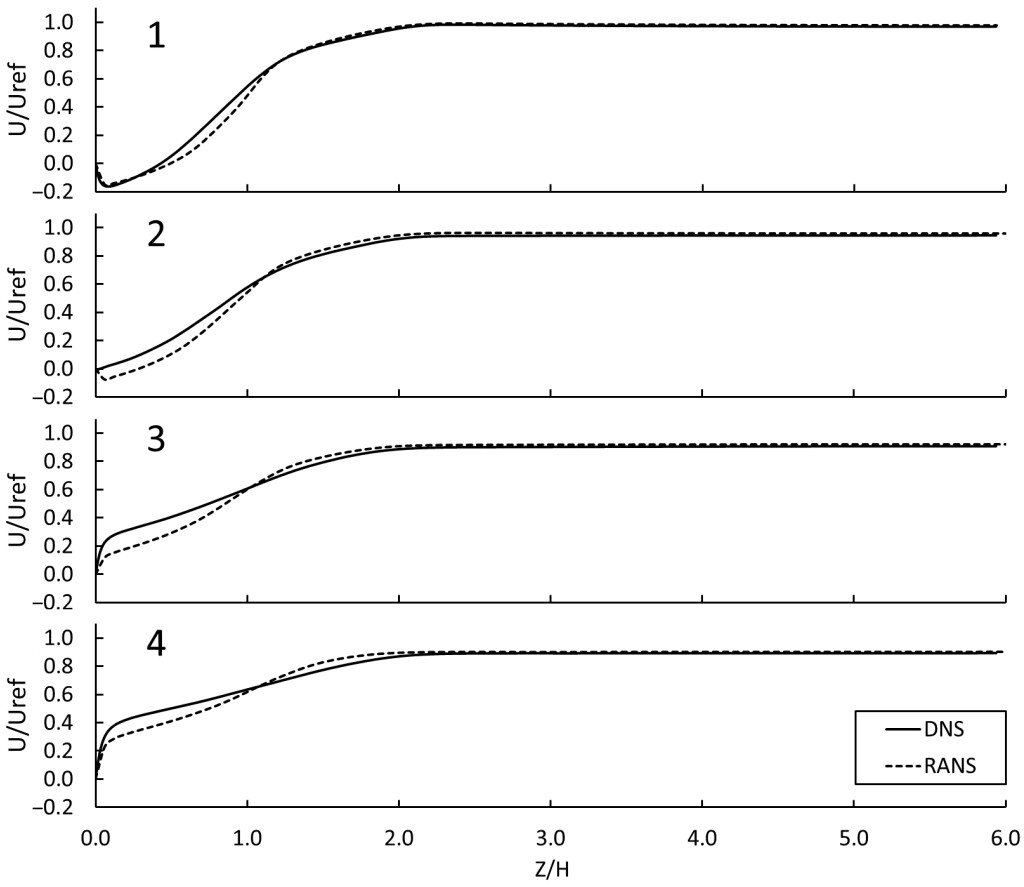

**Figure 12.** Vertical distributions of axial velocity for the *k-ω* SST model in comparison with DNS data by Le and Moin [38] at several positions downstream of the step normalized by step height H. (1): X/H = 4, (2): X/H = 6, (3): X/H = 10, (4): X/H = 15.

### 6. Results

The findings presented above demonstrate a reduction in the length of the separation region caused by the knitted wire mesh. The following numerical results were obtained to further investigate the aerodynamic effect of the knitted wire mesh on flow separation and to examine the impact of the number of rows, as well as the knitted wire mesh geometry.

### 6.1. Effect on Flow Separation

Figures 13–15 show the simulated flow separation affected by the knitted wire mesh. As seen, at spanwise locations in between two stitches, a recirculation area is formed at the upstream end of the ramp. The recirculation area is followed by an area of reattached flow that eventually separates again at the downstream end of the ramp, forming a separation bubble (Figure 13). The reattachment length normalized by $H_2$ at the plane R was 5.01. This represents a reduction of 30.7% compared to the clean ramp. The structure of the separation area varies over the span of the ramp, as indicated by the $y^+$ distribution on the channel bottom wall (Figure 13).

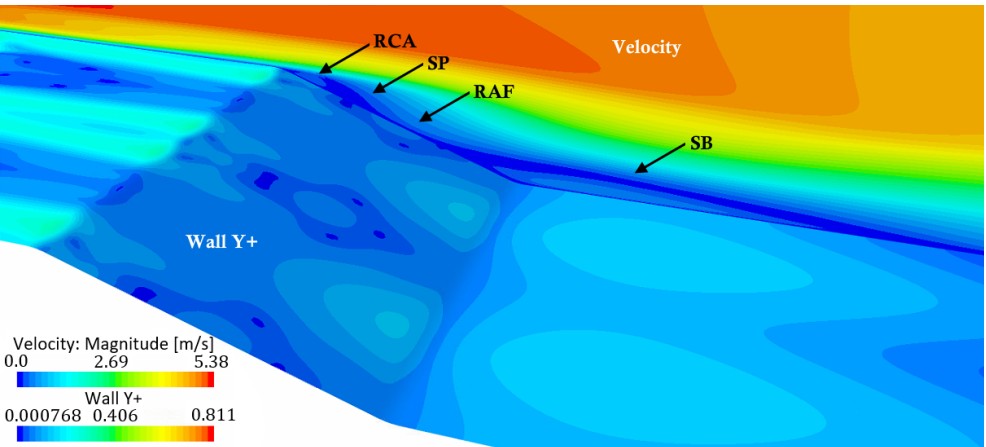

**Figure 13.** Simulated flow separation affected by the knitted wire mesh with two rows. The velocity is shown on the plane R in between two stitches. On the bottom channel walls, the y+ value is presented. SB denotes a separation bubble, SP a stagnation point, RCA a recirculation area and RAF a reattached flow.

Figure 14 displays the velocity distribution in the streamwise direction on plane R. As seen, a recirculation area exists on the upstream end of the ramp, as well as on the downstream end.

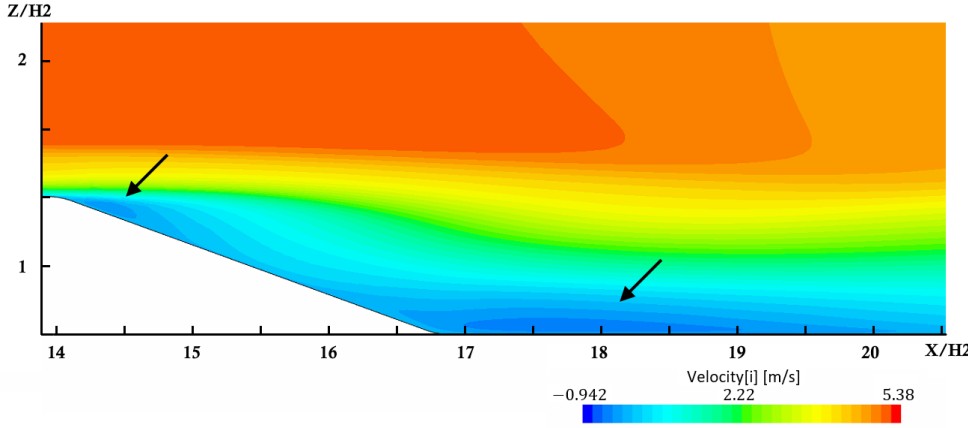

**Figure 14.** Simulated flow separation affected by the knitted wire mesh with two rows. The velocity in the horizontal direction is shown on the plane R in between two stitches. The arrows indicate recirculation areas.

At locations on Plane S, downstream of the stitches, the flow separates at the upstream edge of the ramp, forming a separation bubble (Figure 15). The reattachment length normalized by $H_2$ at the plane S was 6.71. This resulted in a reduction of 7.2% compared to the clean ramp.

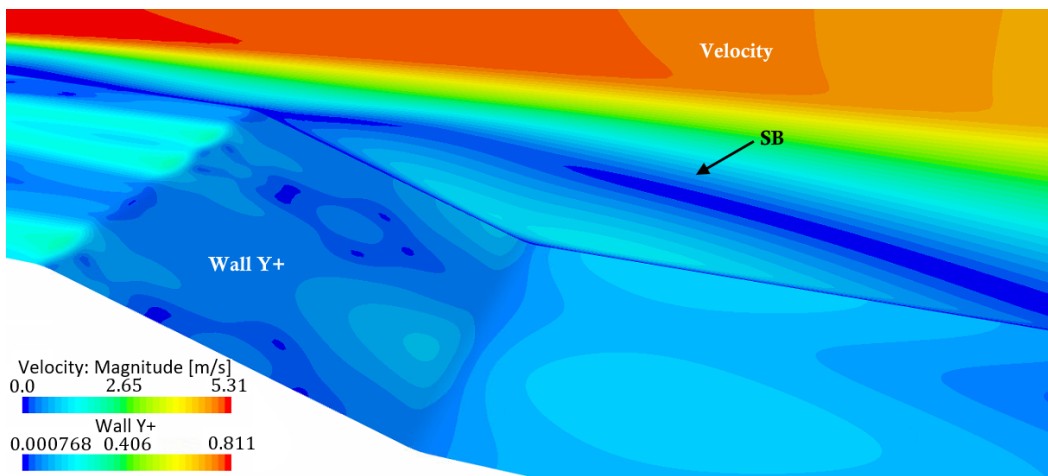

**Figure 15.** Simulated flow separation affected by the knitted wire mesh with two rows. The velocity is shown on the symmetry plane S of a stitch. On the bottom channel walls, the $y^+$ value is presented. SB denotes a separation bubble.

Figure 16 presents the simulated pressure distribution on the channel bottom. As seen, there are areas of low pressure on the knitted wires where the stitches meet. At the locations where the wires contacted the channel ground, positive pressure regions were present, with the flow impinging on the wires. The wake of the wires was associated with an area of low pressure, extending until the wake reached the upper edge of the ramp.

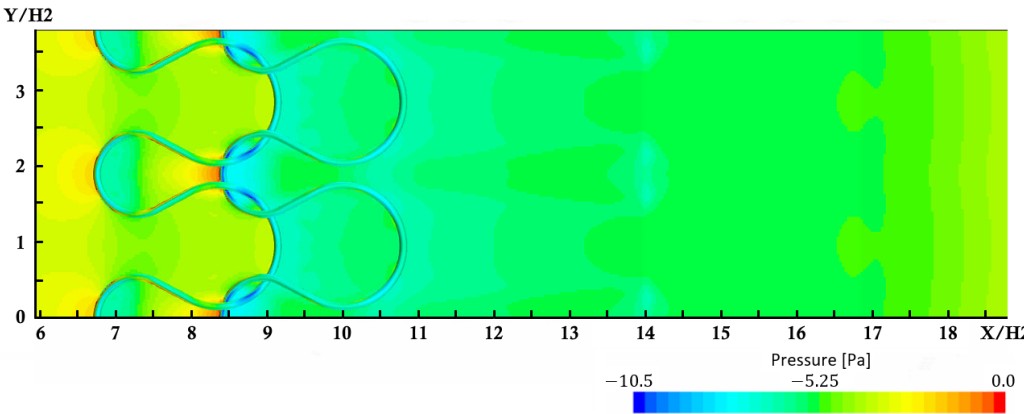

**Figure 16.** Simulated pressure distribution on the channel wall. The ramp edge is located at $X/H_2 = 14$.

### 6.2. Induced Vortices

The simulations exhibit several streamwise vortices in the wake of the knitted wire mesh structure. Figure 17 displays the isosurface of the Q-criterion around the knitted wire mesh. Several streamwise structures are present that originate from the stitches. For the downstream row, each stitch generates two long streamwise structures reaching into the separated flow at the backward-facing ramp. The formation of the streamwise vortex structures was tracked by observing the streamlines of the flow over the knitted wire mesh. Figure 18 displays these streamlines. The figure indicates that the obstruction of the wires forces the fluid to follow the curve of the wires (see denotation 1 in Figure 18). The streamlines denoted by 1 separate into two parts. One part runs over the wires and carries high-momentum fluid. The second part runs under the wires until the downstream end of the knitted wire mesh is reached. There, the flow separates into two streams. One part is directed under the wire and to the outside of the stitch. Then, following the curvature of the wire, the fluid flows to the centerline of the stitch. The other part is directed to the inside of the stitch. From there, the fluid flows over the wire and towards the lateral

end of the stitch (see denotation 2 in Figure 18). Some of the streamlines run inside of the stitches in the proximity of the lateral sides and merge with the other streamlines (see denotation 3 in Figure 18). All streams merge downstream of the stitch and interlace into a streamwise vortex with a downward rotation at the lateral side of the stitch (see denotation 4 in Figure 18). The streamlines of the streamwise vortices elongate into the separated shear layer at the ramp (not shown in Figure 18, but visible in Figure 17).

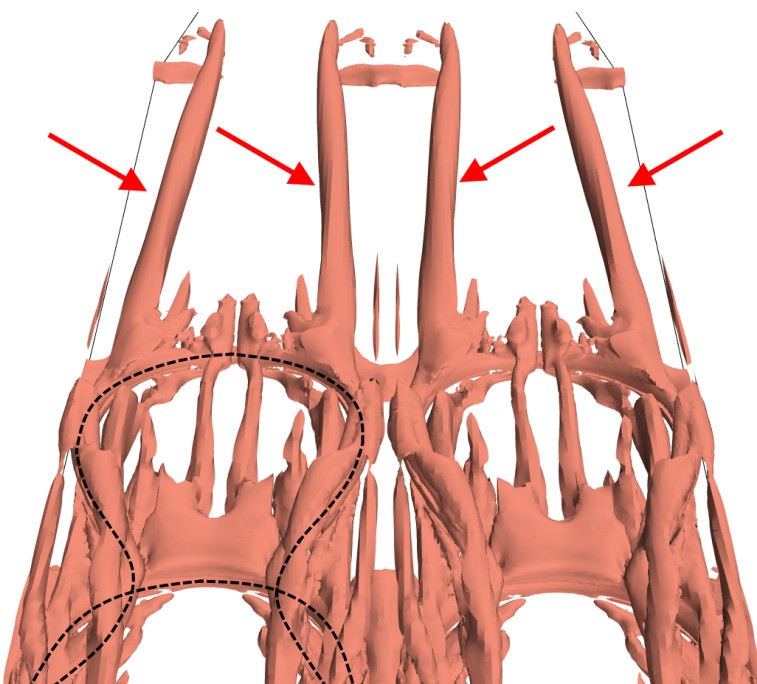

**Figure 17.** Isosurface of the Q-criterion around the knitted wire mesh. The red arrows mark the elongated streamwise vortex structures. The dashed lines indicate the knitted wires. The isovalue is set to 100,000 $1/s^2$.

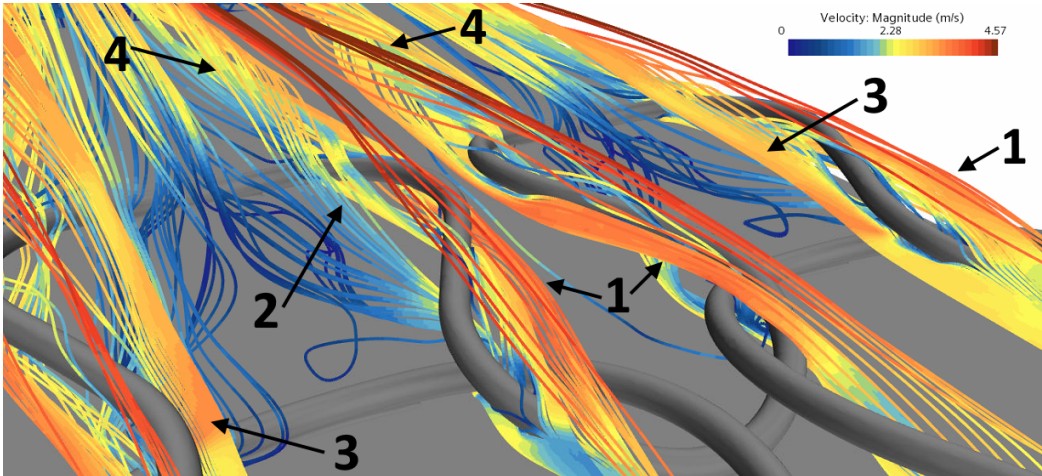

**Figure 18.** Streamlines around the knitted wire mesh with two rows. (1) Streamlines follow the curvature of the wire and extend from under the stitch towards the centerline. (2) Streamlines separate from 1, running inside of the stitch and over the wire towards the lateral outside of the stitch. (3) Streamlines running over the inside of the stitches. (4) Streamwise vortices emerging from 1, 2 and 3.

The streamwise vortices were related to the generation of turbulent kinetic energy. Figure 19 displays the simulated Q-criterion in comparison with the turbulent kinetic energy, the vorticity and the axial velocity in the wake of the knitted wire mesh. Elevated

levels of turbulent energy were observed above the vortices that distributed the turbulence and transported it towards the wall. The turbulent energy was caused by the increased vorticity present in the shear layers between the wake and the free stream flow.

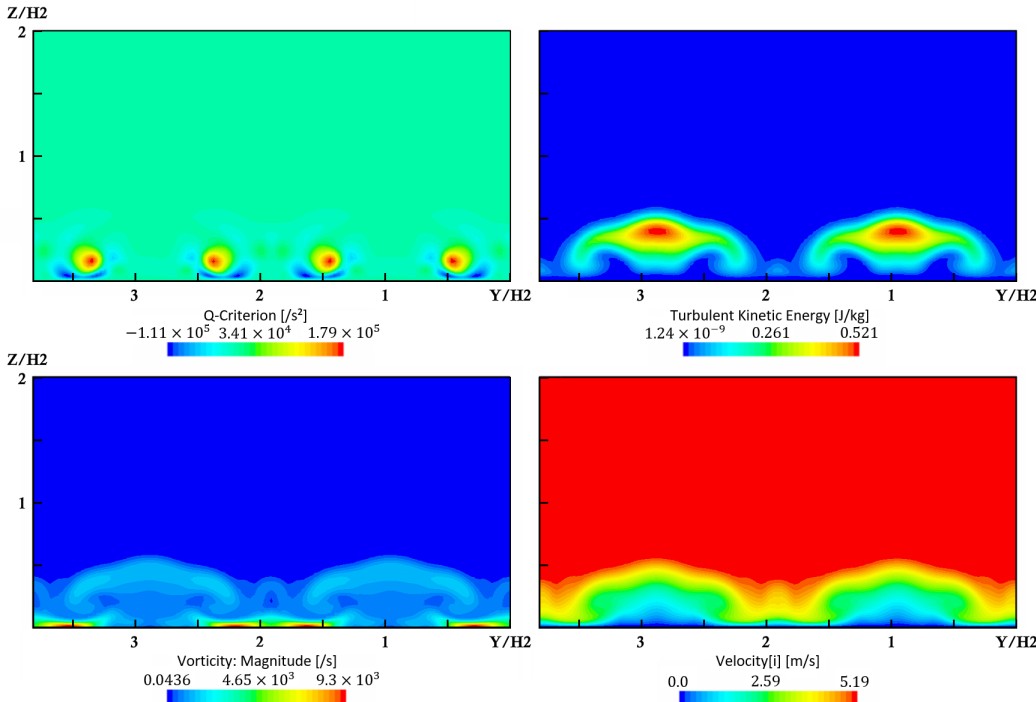

**Figure 19.** Comparison of simulated Q-criterion, turbulent kinetic energy, vorticity and axial velocity on a cross-section in the wake of the knitted wire mesh at $X/H_2 = 13.17$.

### 6.3. Effect of the Number of Rows

Table 7 lists the simulated reattachment lengths and reductions for the number of knitted rows ranging from one to five. The listed reattachment length varies with the number of rows. The model containing one row exhibited the lowest reattachment length reduction of 2.4%. For the number of rows between one and four, the reattachment length reduction increased with the number of rows. The model with four rows featured the greatest reduction of 25.7%. For five rows, the reattachment length reduction decreased to 12.3%. The correlation between reattachment length and number of rows can be described by a quadratic relationship with a coefficient of correlation $R^2$ of 0.985. A potential explanation for this correlation is given in Section 7.

**Table 7.** Calculated reattachment lengths for different numbers of rows of the type A wire mesh.

| Number of Rows | Mean Reattachment Length in mm | Mean Reattachment Length Clean in mm | Mean Reattachment Length Reduction |
|---|---|---|---|
| 1 | 70.5 | 72.3 | 2.4% |
| 2 | 58.5 | 72.3 | 19.0% |
| 3 | 54.6 | 72.3 | 24.5% |
| 4 | 53.7 | 72.3 | 25.7% |
| 5 | 63.7 | 72.3 | 12.3% |

Figure 20 displays the calculated Q-criterion and the velocity magnitude on a cross-section upstream of the backward-facing ramp for different number of rows. As seen, the intensity of the streamwise vortices vanishes with an increasing number of rows (Figure 20a–e). Simultaneously, the velocity of the fluid increases above the knitted wire mesh as well as in between the stitches (Figure 20f–j). For an increasing number of rows, the low-momentum regions in the wake of the stitches grow along with the general size

of the wake. For four rows, a recirculation area emerges in the wake of the stitches on the symmetry plane S (Figure 20i). For five rows, the recirculation area is positioned further upstream and almost vanishes on the cross-section (Figure 20j).

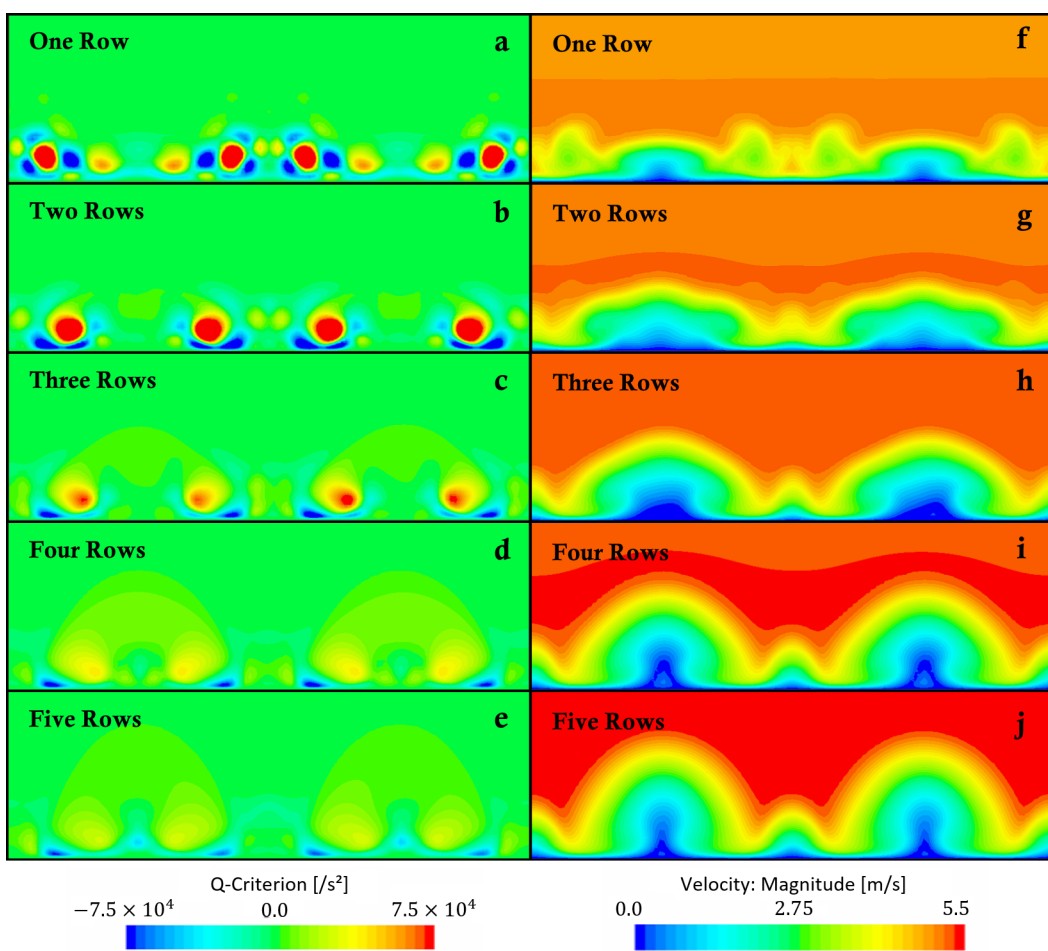

**Figure 20.** Q-criterion and velocity for between one and five knitted wire rows at $X/H_2 = 11.5$, 13.17, 14.83, 16.5 and 18.17, respectively. The X values represent the same distance downstream in the wake of the different wire meshes.

### 6.4. Effect of the Type B Knitted Wire Mesh Geometry

Figure 21 displays the velocity distribution surrounding the ramp for type A and type B knitted wire mesh with two rows. The simulations of type A exhibited a recirculation area at the upstream end of the ramp. The results for type B do not feature a delay of the main separation. The mean reattachment length for type B of 63.7 mm represents a reduction of 11.9%. Type A caused a reduction of 19.0%. Upstream of the ramp, type A features a thinner boundary layer than type B (Figure 21).

Figure 22 presents a comparison of the Q-criterion on a cross-section in the wake of the different kinds of knitted wire meshes. The simulations of the type B mesh yielded four streamwise vortices generated at each stitch. However, all vortices emerging from the type B mesh were weaker than the two vortices generated at the knitted type A mesh.

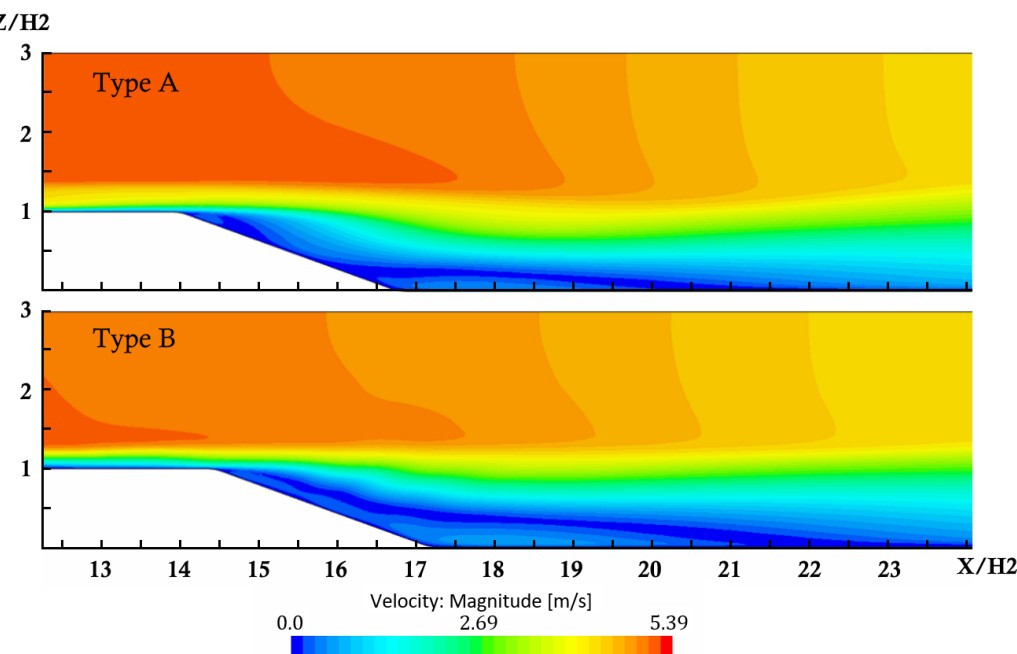

**Figure 21.** Simulated velocity field at the backward-facing ramp for type A and type B knitted wire meshes.

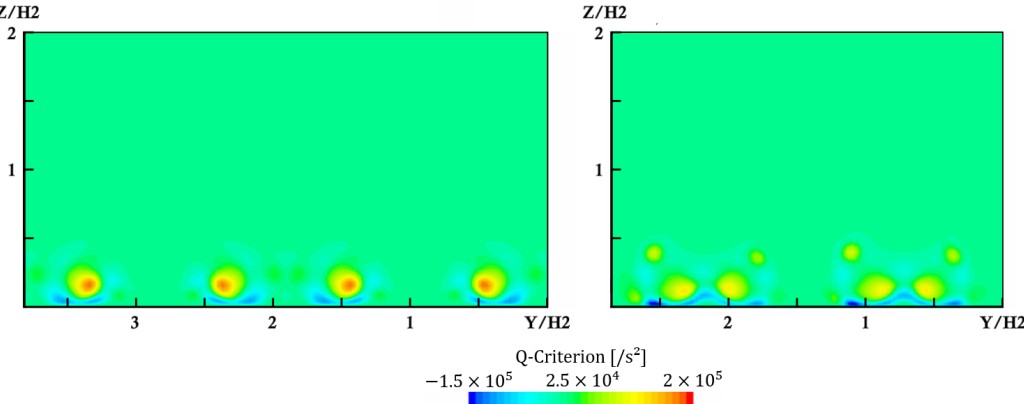

**Figure 22.** Simulated Q-criterion at the same downstream distance in the wake of the knitted wire meshes. **Left**: type A at $X/H_2 = 13.17$, **right**: type B at $X/H_2 = 10.9$. The X values represent the same distance downstream in the wake of the different wire meshes.

## 7. Discussion

The results provide numerical evidence that knitted wire meshes can affect the flow on a flat plate in a beneficial way to reduce flow separation at a backward-facing ramp. As explained as follows, the passive flow control mechanism can be attributed to the streamwise vortex pairs that evolved from each stitch and into the separated shear layer behind the backward-facing ramp.

The flow investigated in this work separated at the edge of the clean ramp due to an adverse pressure gradient. An adverse pressure gradient is a consequence of the change of the geometry at the ramp. The separated flow exhibited a shear layer and eventually reattached downstream of the ramp, forming a separation bubble. Inside, a recirculation area was formed by a vortex. This backward-facing ramp flow was also investigated and characterized by Lim and Lyu, as well as Cuvier et al. [42,43]. The described flow characteristics of the backward-facing ramp are similar to the backward-facing step flow, which represents a 90° ramp. An extensive review on the literature concerning the fluid dynamics and flow control of the backward-facing step flow can be found in [44].

The reduction of the observed separation was associated with increased mixing caused by the noticed counter-rotating streamwise vortices. The vortices exchanged low-momentum fluid close to the wall with high-momentum fluid from the free stream. Due to the large scale of the vortices, this exchange was not homogeneous. In the areas where the motion of the vortices was oriented towards the wall, a thinned boundary layer was observed. In the areas where the motion of the vortices was oriented away from the wall, the boundary layer was thickened. This effect can be seen in Figures 13 and 15. The increased momentum close to the wall provided an enhanced resistance against the adverse pressure gradient and, thus, delayed the separation and reduced the reattachment length.

A similar flow control mechanism based on streamwise vortices is known to occur with vortex generators. A reduction of flow separation was reported for a variety of vortex generators such as wedges, tabs, or cubes. An extensive review on the literature concerning flow control using vortex generators as well as the introduced counter-rotating streamwise vortices is given in [45]. Ma et al. carried out particle image velocimetry measurements for wedge-type vortex generators upstream of a backward-facing step, and they also reported a reduction of the reattachment length [46]. Park et al. investigated the effect of tabs at a backward-facing step experimentally for a Reynolds number of 24,000. They reported streamwise vortices behind the tabs and a decreased reattachment length [47]. Shinde et al. carried out LES and URANS to investigate the flow control of cubes ahead of a backward-facing ramp. They also reported a reduced separation bubble due to the energy exchange introduced by the cube-shaped vortex generators [48]. However, the shape of these vortex generators differs from the stitches of the knitted wires used here. This is especially true for type A (compare Figure 1).

As described in the introduction, streamwise vortices were also reported for dimples. These vortices emerge as a consequence of the flow over the edges of the dimple and inside of the indentation. The vortex mechanics are complex, including secondary flows, vortex shedding, and side-alternating tornado-like vortices [17,18]. The pair of streamwise vortices reported in [18] resemble the vortices found in this investigation. However, the geometric similarity between dimples and the stitches of the knitted wire meshes is limited, especially for type B (Figure 1).

The vorticity in the streamwise vortices behind the knitted wire mesh can be ascribed to the curved surface of the stitches. The wires follow a curved three-dimensional path visible in Figure 1. The observed streamlines split, followed the curvature of the wires, and reunited to form the long streamwise vortex pairs. The vortex pairs originated on the three-dimensionally curved part of the wire at the downstream end of the stitch. Hence, the generation of the streamwise vortices differed from those observed for vortex generators and dimples. The observed aerodynamic mechanism implies that the effect on flow separation can be adjusted via the stitch width, as well as the three-dimensionally curved path of the wire.

The importance of the geometry is underlined by the results obtained using the type B knitted wire mesh. For this type, a smaller effect on flow separation was observed. This could be attributed to lower mixing due to weaker observed induced streamwise vortices (compare the boundary layers upstream of the ramp in Figure 21). Furthermore, an upwash effect of the type B stitch geometry, which transported momentum away from the wall, was observed. Poorer flow separation control due to the upwash effect was also reported by Lim and Lyu for a sweeping jet actuator upstream of a backward-facing ramp [42].

The comparison of the effect of the number of rows revealed a nonlinear correlation between reattachment length and number of rows, indicating interactions between several effects. For an increase of the number of rows, additional geometric obstructions as well as friction losses prevailed inside of the knitted wire mesh. This possibly caused the observed reduced vortex strength as the number of rows increased (Figure 20a–e). A reduction of the vortex strength was reported to lead to an increase of the reattachment length [49]. Simultaneously, the effective blockage ratio increased with the number of rows. This increased the momentum of the fluid above the knitted wire mesh, which flowed inside

the streamwise vortices (Figures 18 and 20f–j). The increased momentum of the fluid close to the wall provides increased resistance against flow separation. Moreover, the wake of the stitches was enhanced with additional rows, which explains the recirculation area upstream of the ramp for four and five rows (Figure 20i,j). A shift of this recirculation area also affected the flow separation and reattachment. A combination of these mechanisms led to the varying reattachment lengths for the different numbers of rows.

A different effect was reported for double-row vortex generators [5,50]. A doublet or double-row arrangement of vortex generators increases the strength of the streamwise vortices and thus improves the separation reduction [5,45]. Nevertheless, a decrease of the streamwise vortex strength was observed here for an increasing number of rows. However, for the numbers of rows between one and three, a similarly beneficial effect on the flow separation was observed for the knitted wire mesh.

Permeability effects of the knitted wire mesh were not observed in this study. This was due to the large scale of the wire mesh used here. As a result, the fluid streamed almost freely in some areas and was blocked in other areas, e.g., downstream of the wires. This can be seen in Figure 18. To investigate permeability effects, the knitted mesh needed to be downscaled and stacked in multiple layers. Hence, the results of this study were not comparable to the experiments with small scale woven lattice structures of Szyniszewski et al. [13].

Isomoto and Honami published experimental results for large-scale single-layer lattice structures upstream of a backward-facing step. They reported increased turbulence intensity due to the rectangular grids that reduced the reattachment length [51]. This indicates a difference in aerodynamic effects between knitted wire meshes and rectangular grids. In contrast to the findings of Isomoto and Honami, the curved stitches of the knitted wires used here produced coherent streamwise vortices. However, for both cases, a beneficial effect on flow separation was observed.

The aerodynamic mechanism of the knitted wire meshes prompted us to propose them for passive flow control as a potential alternative to dimples or vortex generators. Knitted wire meshes could be mounted on airfoils, turbine blades or vehicles to delay and reduce flow separation. In future studies, we will investigate the effect of different types of wire meshes to delay or reduce flow separation on the blades of the NREL 5 MW wind turbine. Other potential applications include mixing or cooling problems where the increased mixing associated with the streamwise vortices can be taken advantage of.

However, all potential real-world applications entail further technical challenges. One is associated with a durable fixture of the knitted wires without altering the geometry or disturbing the flow. Hence, gluing or clamping may not be trivial. Another challenge emerges when applied outside, where fouling or soiling of the stitches increases over time and deteriorates the aerodynamic mechanism.

## 8. Conclusions

Comparative RANS CFD simulations were carried out to investigate the effect of knitted wire meshes on the flow separation on a backward-facing ramp at a Reynolds number of 3000. The $k$-$\omega$ SST model exhibited a significant reduction of 19.0% of the mean reattachment length for the model containing two knitted wire rows. Additionally, at planes in between the stitches of the knitted wire mesh, the separation was reattached, before separating again at the downstream end of the ramp. The reduced reattachment lengths were due to strong streamwise counter-rotating vortices generated by the stitches of the knitted wire mesh.

An analysis of the effect of the number of rows indicated a quadratic correlation with a maximum reattachment length reduction of 25.7% for four rows. A comparison with a different type of knitted wire mesh revealed an increase of the reattachment length, emphasizing the importance of the knitted mesh geometry.

The grid sensitivity analysis yielded insignificant discretization errors on the medium grid. The turbulence sensitivity analysis showed that the $k$-$\omega$ SST model predicted the flow

field in accordance with several other RANS turbulence models. These results justified our confidence in the existence of a significant beneficial effect of the knitted wire mesh on flow separation that is attributed to the streamwise vortices.

The qualitatively significant results of this numerical investigation call for a systematic analysis using advanced methods, such as LES techniques, and a subsequent validation against experimental results.

**Author Contributions:** Conceptualization, J.H.H., F.-J.P. and B.O.e.M.; methodology, J.H.H. and B.O.e.M.; software, J.H.H.; formal analysis, J.H.H.; investigation, H.D. and J.H.H.; resources, F.-J.P.; data curation, H.D.; writing—original draft preparation, J.H.H.; writing—review and editing, H.D., F.-J.P. and B.O.e.M.; visualization, J.H.H.; supervision, F.-J.P. and B.O.e.M.; project administration, J.H.H.; funding acquisition, J.H.H. and F.-J.P. All authors have read and agreed to the published version of the manuscript.

**Funding:** This research was funded by the Federal Ministry for Economic Affairs and Climate Action via the AiF Project under the grant number KK5020302CL0 within the ZIM program.

**Institutional Review Board Statement:** Not applicable.

**Informed Consent Statement:** Not applicable.

**Data Availability Statement:** Not applicable.

**Acknowledgments:** We are grateful to Robert Ruf from Eloona GmbH for provision of knitted wire mesh CAD models. We are grateful to Thomas Erling Schellin for review of the manuscript. We acknowledge support by the Open Access Publication Fund of the Westphalian University.

**Conflicts of Interest:** The authors declare no conflict of interest. The funders had no role in the design of the study; in the collection, analyses, or interpretation of data; in the writing of the manuscript; or in the decision to publish the results.

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
