# Peer review of "Aerodynamic Effects of Knitted Wire Meshes—CFD Simulations of the Flow Field and Influence on the Flow Separation of a Backward-Facing Ramp"

_fluids, doi:10.3390/fluids7120370_

Round 1

Reviewer 1 Report

The paper is a clear description of a CFD analysis performed on two different type of knitted wire meshes by using the same solver with several different turbulence models. 

The scientific soudness of the work  must be improved.

The work  does not present any validation of the approach used in solving the proposed flowfield  or for a flowfield as close as possible to it .

Even if the grid sensivity analysis is performed correctly and also other metrics effect are accounted for,  the overal analysis still remain superficial.

The scientific output of the work is an evaluation of the drag reduction  and of the reattachment lenght obtained with different  RANS models and for a single (and unknown) value of the flow velocity.  The results obtained are not conclusive and other sources of numerical or experimental data are not presented/available for comparison.

-  The reviewer suggests to solve at least the baseline flow over  the ramp   for a case available in the literature  ( e.g. Ref 39) by using the different turbulent models adopted in present study.

-  Moreover, aside the turbulence models used, Spalart Allmaras model without any wall or damping function should be tested. In fact the authors claimed that the  y+  < 1 was set for all the simulations. This setting is  not correct for turbulence models that use wall function as k-w  and k-e models, or damping functions as for the low Re k-e model.

- A deeper analysis of the turbulent  field and of the vortex structures must be performed for a better caracterization of the effects  produced by the two kind of wire meshes, with the aim of identifying the geometric parameters of the wire mesh  that positively affect the ramp flowfield.

- Finally, A DES or VLES approach may be more appropriate for capturing the complex interactions between vortex generated by the wire mesh and the wall structures.

Reviewer 2 Report

See file.

Reviewer 3 Report

The authors have numerically analysed the Aerodynamic Effects of Knitted Wire Meshes-Flow Field and their Influence on the Flow Separation of a Backward-Facing Ramp.

The manuscript is well written and presents its novelty and importance with simulation methodology.

The minor suggestions are as follows, which may help improve the manuscript.

·       The reduction in % needs to be mentioned in the abstract.

·       Keywords need to be reduced to five only.

·       Name the equation (5) and (6).

·       The working fluid used for the simulation needs to be mentioned along with considered constant properties.

·        The author has mentioned that simulation was performed for Re=3000; What is the justification for the selection of Re with its importance and field application?

·       What was the reason for not presenting the pressure plots, as it is also an essential parameter in the flow separation study?

Round 2

Reviewer 1 Report

None.

Author Response

We thank the reviewer.
